# Single-Port Coherent Perfect Loss in a Photonic Crystal Nanobeam Resonator

**DOI:** 10.3390/nano11123457

**Published:** 2021-12-20

**Authors:** Jihoon Choi, Heeso Noh

**Affiliations:** Department of Physics, Kookmin University, Seoul 02707, Korea; aries153@kookmin.ac.kr

**Keywords:** coherent perfect absorption, photonic crystal, single-port, PMC boundary

## Abstract

We numerically demonstrated single-port coherent perfect loss (CPL) with a Fabry–Perot resonator in a photonic crystal (PC) nanobeam by using a perfect magnetic conductor (PMC)-like boundary. The CPL mode with even symmetry can be reduced to a single-port CPL when a PMC boundary is applied. The boundary which acts like a PMC boundary, here known as a PMC-like boundary, and can be realized by adjusting the phase shift of the reflection from the PC when the wavelength of the light is within the photonic bandgap wavelength range. We designed and optimized simple Fabry–Perot resonator and coupler in nanobeam to get the PMC-like boundary. To satisfy the loss condition in CPL, we controlled the coupling loss in the resonator by modifying the lattice constant of the PC used for coupling. By optimizing the coupling loss, we achieved zero reflection (CPL) in a single port with a PMC-like boundary.

## 1. Introduction

Absorption of light is an important property for control of traveling of light and removing the desired information from light. In recent years, many studies have been implemented to increase the light absorption efficiency based on metal nanostructure, graphene and multilayer structure [1,2,3,4]. In optical circuits, absorption was used in on-chip devices for controlling light in a certain route such as optical switch and modulator [5,6,7]. Use of a time-reversed laser allow perfect absorption, also known as coherent perfect absorption (CPA) [8,9,10]. CPA has been studied theoretically and experimentally because it uses a novel concept to achieve perfect absorption. In various fields involving, for example, surface plasmons and defected photonic crystal resonators [11,12,13,14]. Furthermore, CPA-lasing was studied using PT-symmetric optical structure [15,16]. Recently, an optical logic gate and a refractive index sensor based on CPA have been reported [17,18]. The CPA can be explained using a simple Fabry–Perot resonator. In such a resonator with an absorbing medium, two lights with equal intensities incident on both sides are partly reflected and partly transmitted at the boundary of the resonator. By controlling the relative phase between the two incident light rays, one can achieve destructive interference between the reflected light and transmitted light. The destructive interference results in the complete disappearance of the outgoing light and only the incident light remains. Because CPA is time-reversed lasing, the incident light is in resonance and is completely absorbed inside the resonator.

CPA requires accurate phase control between the two incident light rays. If a resonator has a symmetric structure, such as the symmetric Fabry–Perot resonator, the resonance mode in the resonator should have even or odd symmetry. Therefore, the relative phase difference between the two incident waves should be 2nπ or (2n+1)π(n=0,1,2,…), depending on the symmetry, where *n* is an integer. Figure 1 shows an example of simple CPA with a symmetric Fabry–Perot resonator. The resonators with even and odd symmetries are depicted in Figure 1a,b, respectively. To obtain destructive interference between the outgoing waves, the two incident waves should be in phase for even symmetry, or they should have a phase difference of (2n+1)π for odd symmetry.

In real experiments, one incident wave is preferred to two incident waves due to the control of light. To achieve single-port (one incident wave) CPA, we must use a boundary such as perfect electric conductor (PEC) or perfect magnetic conductor (PMC) to maintain symmetry. Figure 1c,d depicts single-port CPA with PEC and PMC boundaries, respectively. At the PEC boundary, the electric field component parallel to the boundary and the magnetic field component normal to the boundary are zero ($H_{⊥}=0$ and $E_{\|}=0$), in which the resonant modes have odd symmetry, while the resonant modes have even symmetry with a PMC boundary ($E_{⊥}=0$ and $H_{\|}=0$). Therefore, CPA with odd symmetric and even symmetric resonators correspond to resonators with PEC and PMC boundaries, respectively.

Sucheng Li et al. reported CPA achieved with PMC boundary surfaces [19]. This is because metasurfaces using a high-impedance surface can cause magnetic resonance; therefore, a PMC boundary could be realized. In this PMC boundary condition, CPA was obtained in microwaves. However, achieving CPA with a PMC boundary on a chip is challenging due to the difficulty of fabricating a PMC metasurface.

The boundary which behaves as PMC or PEC can be realized with photonic crystals (PCs) on a chip. We named this boundary the PMC(PEC)-like boundary. A PC affects the transmission and reflection of light in a medium and can be prepared using a periodic array of two or more media with different refractive indices. In particular, the PC can act as a perfect reflector in a specific wavelength range called a photonic band gap. A simple example of a PC is the Bragg reflector, which is a one-dimensional PC that can perfectly reflect incident light [20,21,22]. Owing to their outstanding reflection properties, PCs have been used in various optical devices such as resonators, waveguides, filters, absorbers, and polarization selectors [23,24,25,26,27,28]. Recently, the use of PC resonators in nanobeams has been studied and demonstrated experimentally, such as nanobeam lasers, programmable nanobeam resonators, and coupled nanobeam resonators [29,30,31]. In a nanobeam, a PC resonator can be designed easily. The quality factor (Q factor) of the PC resonator can be on the order of $10^{9}$ [32]. At the boundary of the PC, a phase shift from −π to π can be introduced for the reflected light in the photonic band gap spectral region [33]. Therefore, not only can a PEC-like boundary be realized according to the wavelength in a photonic band gap of a PC but also a PMC-like boundary.

In this study, we numerically demonstrate coherent perfect loss (CPL) based on CPA with a single port with a PMC-like boundary using a PC. The only difference between CPA and CPL is that absorption is changed into loss. We selected a PMC-boundary to implement CPL with even symmetry and designed an optimized PMC-like boundary by modifying the lattice constant and hole radius of the PC in a nanobeam. We created a simple Fabry–Perot resonator by introducing a PMC-like boundary on one side of the resonator and a PC on the other side, which was used as a coupler. Additionally, to achieve CPL, the loss in the resonator should be controlled, which can be achieved by modifying the lattice constant of the PC used for coupling. After matching the CPL condition, we observed perfect loss in the resonator with a PMC-like boundary.

## 2. Simulation

To realize the PMC-like boundary, we designed a 3-dimensional PC nanobeam on a silicon-on-insulator, which consisted of a 220 nm thick silicon layer on a silicon dioxide substrate. The wavelength of the source light was 1550 nm, which is a commonly used wavelength in the field of telecommunications (Figure 2a). At this wavelength, the refractive indices of silicon and silicon dioxide are 3.48 and 1.44, respectively [34,35]. We used the TE mode ($E_{y}$) for an incident wave, a PC with lattice constant $a_{L}=400$ nm, a ratio of air-hole radius to lattice constant ($r_{L}/a_{L}$) of 0.35, and a nanobeam width of 500 nm. We calculated the photonic band gap using the finite element method (COMSOL Multiphysics) in three-dimension. The wavelength range of the photonic bandgap was 1200–1560 nm.

### 2.1. PMC-like Boundary

To determine the PMC-like boundary, we compared the electric field distribution at the resonance wavelength between the PC serving as the PMC-like boundary and the PMC boundary. The simulation configuration and electric field distribution of the PMC-like boundary using a PC and the PMC boundary are depicted in Figure 2. All simulations are performed in three dimensions. We used a simple Fabry–Perot cavity as a resonator. The red-dashed lines in Figure 2 are the PMC and PC boundaries. The PC as the PMC-like boundary comprised a one-dimensional array of 10 holes with lattice constant $a_{L}=400$ nm and $r_{L}/a_{L}=0.35$. On the right side of the resonator, there were three holes with $a_{R}=450$ nm and $r_{R}/a_{R}=0.35$, which controlled the transmission of the incident light to the resonance region through appropriate reflection. For the resonator with the PMC boundary, the resonator length is defined as the length from the boundary on the left of the resonator to the left edge of the first hole of the PC used for coupling. For the resonator with the PMC-like boundary, we need to consider an additional length to compensate for the additional phase shift, which occurs because of the penetration of the electric field into the PMC-like boundary [36]. Therefore, the length of the resonator (the length from the right edge of the first hole of the PC used as the PMC-like boundary to the left edge of the first hole of the PC used for coupling) is the sum of the effective length and the length to compensate for the additional phase shift. Figure 2b shows cross-sectional view of the norm of electric field distribution inside the waveguide. The light with a single mode propagating in the nanobeam waveguide has Gaussian distribution. Electric field distributions of resonance modes of both cases are depicted in Figure 2 in xy-plane (c) and in xz-plane (d). In Figure 2c,d, we confirm that the electric field distributions ($E_{norm}$) in the resonator with the PMC boundary and those with the PMC-like boundaries are similar.

We first confirmed that the electric field distributions ($E_{y}$) in the resonator with the PMC boundary and those with the PMC-like boundaries are similar. Next, we calculated the resonance wavelength for various effective lengths. The length of the resonator with the PMC-like boundary is not equal to the resonator length of the resonator with the PMC boundary because of the additional phase shift, but the effective length of the resonator with the PMC-like boundary is the same as the resonator length of the resonator with the PMC boundary. The resonance wavelength is red-shifted when the effective length is longer. This is depicted as the black solid line in Figure 3a for the resonator with the PMC boundary. To define the PMC-like boundary, we must compare the resonance wavelengths at the same resonator length (effective length for the resonator with the PMC-like boundary). When the resonator length of the resonator with the PMC boundary is 1967 nm, the resonance wavelength is 1550 nm. The blue-dotted line in Figure 3a shows the resonance wavelength as a function of the effective length for the resonator with the PMC-like boundary with $r_{t}/a_{L}=0.35$. When the effective length is approximately 2000 nm, the resonance wavelength of approximately 1565 nm for the resonator with the PMC-like boundary is equal to that of the PMC boundary. To make the phase shift at the two boundaries equal for the wavelength of 1550 nm, we must change the resonance wavelength of the resonator with the PMC-like boundary to 1550 nm with the effective length of 1967 nm. Because the phase shift of the reflection at the boundary is related to the refractive index, we can adjust the phase shift by changing the effective refractive index at the boundary. This can be achieved by tapering a hole at the boundary of the PC [37]. A plot of the resonance wavelength versus the tapered hole radius at the PC serving as the PMC-like boundary for an effective length of 1967 nm is depicted in Figure 3b. The resonance wavelength is red shifted as the hole radius increases. For $r_{t}/a_{L}=0.23$, the resonance wavelength is 1550 nm. The red-dashed line in Figure 3a represents $r_{t}/a_{L}=0.23$. At this value, the resonance wavelengths of the resonator with the PMC-like boundary and that with the PMC boundary coincided at the effective length of 1967 nm. Finally, the resonator with the PMC-like boundary is characterized by $a_{L}=400$ nm, $r_{L}/a_{L}=0.35$, $r_{t}/a_{L}=0.23$, and a compensation length of 47.5 nm. Note that for this condition, the PC as the PMC-like boundary on the left side of the resonator behaves as the PMC boundary at a wavelength of 1550 nm.

### 2.2. Single-Port Coherent Perfect Loss

To achieve CPL, the loss in the resonator should be controlled. At the wavelength of 1550 nm, SiO_2_ and Si do not have optical absorption. However, the loss in terms of the CPL includes not only absorption but also coupling loss in the resonator. In our case, the loss in the CPL condition originates from the coupling loss at the PC boundary which is the scattering loss to free space. Even if there is intrinsic absorption in material, the CPL can be achieved when the loss condition is satisfied. The main propagation direction of light is in the *x*-axis (reflection and transmission direction is in the *x*-axis). Therefore, light in all other direction except the *x*-axis is lost. Figure 4a shows a schematic of the CPL simulation. The incident light source propagated only along the −x direction from the right side of the resonator by using the unidirectional incident wave [38]. Detectors located on the right side of the source and the left side of the resonator recorded the intensities (amplitudes of the average Poynting vectors) along the *x* axis to determine the intensities of the reflection and transmission, respectively. The incident light wavelength ranged from 1540 nm to 1560 nm. The PMC-like boundary on the left of the resonator consisted of 10 holes and 1 tapered hole. The length of the resonator was 2014.5 nm. The PC on the right side of the resonator consisted of 3 holes for coupling. Resonance wavelength was 1550 nm in this structure.

For single-port CPL, only the incident light should exist, without the reflected light. Therefore, the intensity in the detector should be zero. At the detector for transmitted light, the intensity was almost zero because the PC in the PMC-like boundary perfectly reflected the light. The intensity reflected from the resonator is depicted in Figure 4b (blue solid line) as the normalized intensity $I/I_{0}$, where $I_{0}$ is the reference intensity. The reference intensity was recorded with the same source without the structure. The minimum normalized intensity $I_{\min}/I_{0}$ is 0.1843 when $a_{R}=450$ nm, because the loss in the resonator is not perfectly matched for CPL. We could control the loss in the resonator by modifying the lattice constant $a_{R}$ of the PC on the right side of the resonator. It is also necessary to modify the resonator length to match the resonance wavelength with the desired wavelength when modifying $a_{R}$. As $a_{R}$ changes from 400 nm to 450 nm, $I_{\min}/I_{0}$ also changes (Figure 4c). When $a_{R}$ is 450 nm, the loss in the resonator is not sufficient to achieve CPL, as mentioned above. Coupling loss from the PC on the right side of the resonator increases as $a_{R}$ is reduced from 450 nm to 420 nm. For $a_{R}=420$ nm, $I_{\min}/I_{0}=0$. This means that the increased loss in the resonator results in a 100% total loss, which occurs because the loss in the resonator satisfies the CPL condition. In this condition, the full width at half maximum of the loss is 2.25 nm. For $a_{R}=420$ nm and a length of resonator of 2021.7 nm, we achieved single-port CPL using the PMC-like boundary.

We confirmed that effect of the number of holes of PC in PMC-like boundary on performance of the CPL. We obtained the intensities of the dip in reflection spectra ($I_{R}$) and the peak in transmission spectra ($I_{T}$) as a function of the number of holes on PMC-like boundary (Figure 5a). As number of holes is increased to 5, $I_{R}$ is rapidly decreased, and it is kept zero for number of holes more than 5 (blue circles in Figure 5a. It is because scattering loss at resonator is changed as the number of holes is changed to 5. As a result, $I_{R}$ is decreased because scattering loss at resonator is matched with CPL condition as number of holes on PMC-like boundary is increased. In case of $I_{T}$, it is decreased rapidly as number of holes is increased to 9 and kept almost 0 when the number of holes is more than 9 (red circles in Figure 5a). Because the penetration of electric field at PC on PMC-like boundary is weaker with increasing number of holes, $I_{T}$ is decreased to 0. The black circles in Figure 5a are the dip in total intensity (sum of intensities of transmission and reflection spectra) spectra ($I_{T+R}$). When number of holes on PMC-like boundary is increased, $I_{T+R}$ is decreased and kept almost 0 for more than 9 of holes.

Figure 5b shows the wavelength of $I_{R}$ and $I_{T}$ between 1450 nm and 1650 nm. As the number of holes is increased to 5, wavelengths of $I_{T}$ (red circles) and $I_{R}$ (blue circles) are blue-shifted. $I_{T+R}$ is also blue shifted to 1550 nm with increasing number of hole. In PC of resonator, penetration depth are almost maintained at more than 9 holes, while in less than 9 holes, more electric field is penetrated to waveguide on the left of PC, i.e., electric field at resonator is coupled with waveguide. As a result of coupling between waveguide and resonator, effective length of resonator are changed. The coupling between resonator and waveguide on the left side becomes weaker with increasing number of holes, which leads to shorter effective length of resonator. Consequently, wavelength of resonance are blue-shifted. As number of holes is more than 9, wavelengths of $I_{T}$, $I_{R}$ and $I_{T+R}$ are close to 1550 nm. Therefore, 10 holes are smallest number of holes for PMC-like boundary. The photonic band gap is affected by temperature because the refractive index of material is varied with temperature [39]. However, the refractive index of silicon used in our simulation is robust from temperature. The refractive index of silicon varies by approximately ±0.01 at ±50 K difference from room temperature at the wavelength of 1550 nm [40], which makes the deviation of resonance wavelength smaller than 5 nm.

## 3. Conclusions

We achieved single-port CPL with a PMC-like boundary by using a PC. For $a_{L}=400$ nm, $r_{L}/a_{L}=0.35$, $r_{t}/a_{L}=0.23$, a PMC-like boundary was achieved by controlling the phase shift of the reflection at the PC boundary. We also designed a resonator in the nanobeam by positioning a PC on the nanobeam. On one side of the nanobeam, the PC nanobeam resonator had a PC that served as a PMC-like boundary; the PC on the other side of the nanobeam acted as a coupler. The CPL condition is satisfied when the lattice constant of the PC for coupling is 420 nm and the length of the resonator is 2021.7 nm. For more 10 of holes of PC, PMC-like boundary shows best performance. Owing to the simple configuration and phase manipulation for single-port CPL, our device has the potential to simplify optical integrated circuits such as optical switches and modulators using CPA.

## Figures and Tables

**Figure 1 nanomaterials-11-03457-f001:**
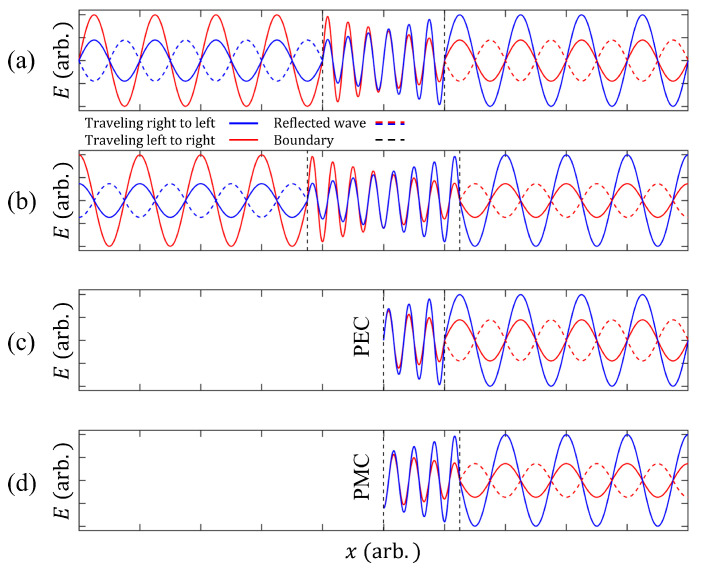
Model of simple CPA with (**a**) odd and (**b**) even symmetry of the resonance mode. The red (blue) curve show traveling waves to the left (right). Solid curves represent incident and transmitted waves; Dashed curves, reflected waves. The black dashed line is the boundary of the resonator. The resonance mode with even symmetry has two incident light rays that are in phase, and the mode with odd symmetry has two incident waves with a phase difference of (2n+1)π (n=0,1,2,…). Single-port resonator with (**c**) PEC and (**d**) PMC boundary. The size of the system with even (odd) symmetric mode can be reduced by using the PEC (PMC) boundary.

**Figure 2 nanomaterials-11-03457-f002:**
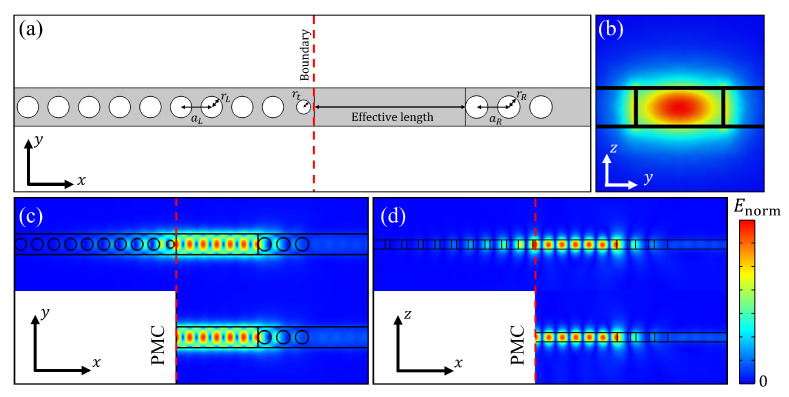
(**a**) Simulation configuration for resonance mode. PC nanobeam resonator consists of nanobeam with 500 nm of width and air holes. The defect in PC nanobeam acts as a simple Fabry–Perot resonator. The role of air holes is PMC-like boundary at left side of resonator and coupler at right side of resonator respectively. Red dashed line is boundary of resonator when PMC and PMC-like boundaries are applied. $a_{L}$ and $a_{R}$ are lattice constants of PC at left side and right side of resonator and $r_{L}$ and $r_{R}$ are radii of holes at left side and right side of resonator, and $r_{t}$ is the radius of tapered hole. (**b**) Cross-sectional view of $E_{norm}$ of wave propagation in waveguide. Norm of the electric field distributions in the nanobeam for the PMC-like (top) and PMC (bottom) boundaries in xy-plane (**c**) and xz-plane (**d**), respectively. The lattice constants of the PC on the left and right sides of the resonator were 400 and 450 nm, respectively, and the hole size of each PC was r/a=0.35.

**Figure 3 nanomaterials-11-03457-f003:**
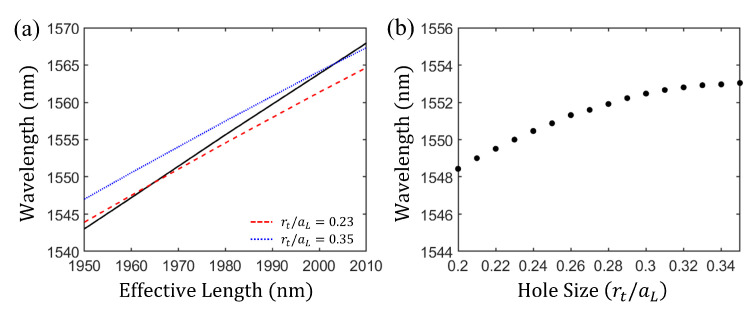
(**a**) Resonance wavelength plotted as a function of the effective length. The black solid line and blue dotted line ($r_{t}/a_{L}=0.35$) represent the resonance wavelength for the PMC boundary and the PC boundary vs. effective length, respectively. In this situation, the effective length of the resonator with the PMC-like boundary is equal to the resonator length of the resonator with the PMC boundary. The red dashed line represents the resonance wavelength for the PC with a tapering hole ($r_{t}/a_{L}=0.23$) vs. effective length. (**b**) Resonance wavelength shift caused by the change in the hole radius ($r_{t}/a_{L}$) of a tapering hole near the boundary of the PC for the effective length of 1967 nm. For the hole size $r_{t}/a_{L}=0.23$, the resonance wavelength was approximately 1550 nm.

**Figure 4 nanomaterials-11-03457-f004:**
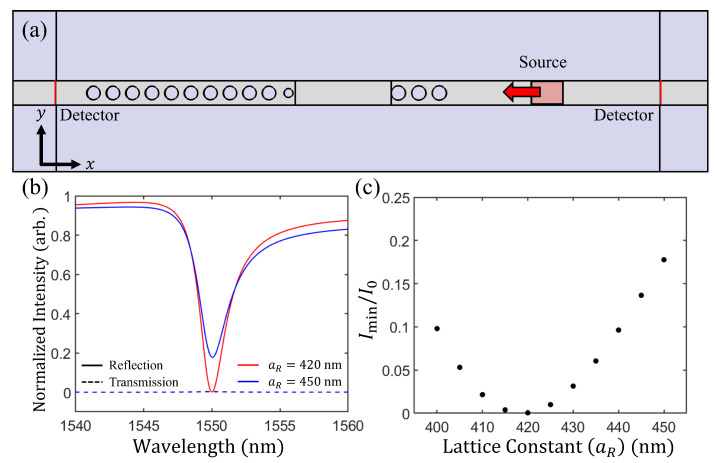
(**a**) Simulation configuration for CPL with the PMC-like boundary. Incident light with the *y*-axis electric field ($E_{y}$) propagated to the resonator. Detectors at the left and right sides collect the Poynting vector of the *x*-axis of the transmitted and reflected light, respectively. (**b**) The intensity of the outgoing light from the resonator. The solid and dashed lines represent the reflected and transmitted lights, respectively. When the lattice constant ($a_{R}$) of the PC on the right side of the resonator is 450 nm (blue line), the loss (in this situation, coupling loss) of the resonator is approximately 82%. The red solid line shows the loss for a lattice constant ($a_{R}$) of 420 nm (red line). The loss reaches 100% at a wavelength of 1550 nm. (**c**) Intensity ratio $I_{\min}/I_{0}$ for the modified lattice constant of the PC acting as a coupler. For the lattice constant of approximately 420 nm, the $I_{\min}/I_{0}$ is approximately zero.

**Figure 5 nanomaterials-11-03457-f005:**
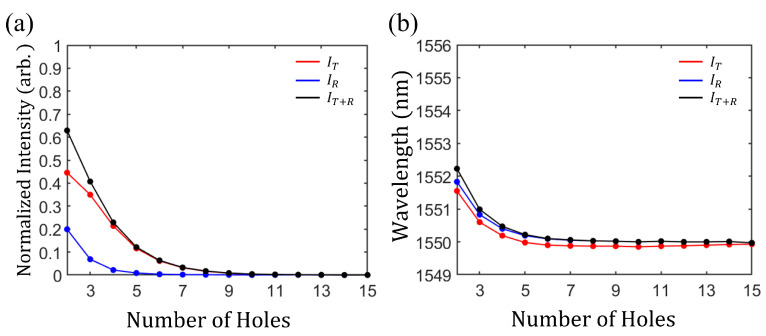
(**a**) Normalized intensity of the peak in transmission spectra ($I_{T}$) and the dip in reflection spectra ($I_{R}$) for various number of holes of PC at PMC-like boundary. $I_{T}$ and $I_{R}$ are depicted as red and blue circles respectively. The black circles show the dip in total intensity spectra ($I_{T+R}$), dip in sum of intensities of transmission and reflection spectra. When number of holes of PC increases, all $I_{T}$, $I_{R}$ and $I_{T+R}$ are decreased. (**b**) Wavelength of $I_{T}$ and $I_{R}$ vs. number of holes. Red and blue circles represent the wavelengths of $I_{T}$ and $I_{R}$. Wavelength of $I_{T+R}$ are depicted as black circles. When number of holes are increased, all wavelengths of $I_{T}$, $I_{R}$ and $I_{T+R}$ are blue shifted to approximately 1550 nm.

## Data Availability

Data sharing not applicable.

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
