# Peer review of "Single-Port Coherent Perfect Loss in a Photonic Crystal Nanobeam Resonator"

_nanomaterials, 2021, doi:10.3390/nano11123457_

Round 1

Reviewer 1 Report

Please see the attached PDF file for the comments.

Author Response

We thank you for the careful and detailed review. We read the review in detail and revised our journal carefully.

• An important feature of coherent perfect absorbers is the possibility of having a broad angle operation, i.e., CPA occurring at different incident angles, ideally from 0 to almost 90 degrees. Can the authors comment on the possibility of having such angle variation with their scheme?
-> In our study, we consider CPA on-chip. In our simulation, light is propagated along a nanobeam with a single mode. Therefore, we do not need to consider the angle of incidence. We modified sentence in paragraph 1 page 4 (Line 113). 
'The light with a single mode propagating in the nanobeam waveguide has gaussian distribution.'

• The authors state and consider the coupling loss at the photonic crystal boundary in opposition to the material loss. How would the results and conclusions of differ in case loss inside the material is present?
-> In our simulation, we used scattering loss instead of material loss. Therefore, even if material loss is increased, CPL can work when the loss condition is satisfied. We added a sentence in paragraph 2 page 5 (Line 155). 
‘Even if there is intrinsic absorption in material, the CPL can be achieved when the loss condition is satisfied.'

• Some papers on the topic need to be discussed in the context of this study, e.g.:
- Teperik, T. V., De Abajo, F. G., Borisov, A. G., Abdelsalam, M., Bartlett, P. N., Sugawara, Y., & Baumberg, J. J. (2008). Omnidirectional absorption in nanostructured metal surfaces. Nature photonics, 2(5), 299-301.
- Longhi, S. (2010). PT-symmetric laser absorber. Physical Review A, 82(3), 031801.
- Liu, N., Mesch, M., Weiss, T., Hentschel, M., & Giessen, H. (2010). Infrared perfect absorber and its application as plasmonic sensor. Nano letters, 10(7), 2342-2348.
- Farhat, M., Yang, M., Ye, Z., & Chen, P. Y. (2020). PT-symmetric absorber-laser enables electromagnetic sensors with unprecedented sensitivity. ACS Photonics, 7(8), 2080-2088. 
-> We added references as ref. [1] and [2] paragraph 1 page 1 (Line 14).
‘In recent years, many studies have been implemented to increase the light absorption efficiency based on metal nanostructure, graphene and multilayer structure.’
And we added sentence and references as ref. [15] and [16] paragraph 1 page 1 (Line 22).
'Also, CPA-lasing was studied using PT-symmetric optical structure.'

Reviewer 2 Report

Manuscript reference number: Nanomaterials - 1478232-

Single-Port Coherent Perfect Loss in a Photonic Crystal Nanobeam Resonator by Jihoon Choi et al.

In this manuscript the authors demonstrate with numerical simulations how to design a single-port coherent perfect loss with a Fabry-Perot resonator in a photonic crystal nanobeam. The paper is rather interesting and well organized. It could be published on this journal after revisions according to the list of the following critical points.

a) Abstract.

 “With appropriate loss, we obtained zero reflection (CPL) in a single port by using a PMC-like boundary”. In the abstract the authors anticipate the main results of the numerical simulations, which are correctly reported in the conclusions with more details. The risk is that the abstract is not clear. We are wondering if the authors may re-write the abstracts just mentioning that in this paper they have studied, analyzed and optimize the structure in order to achieve a nearly zero reflection condition.

b) Figure 1 gives only a general idea of the process. Probably the reference to the real case with real materials etc.. could help. Moreover the “x arb” axis should be better replaced by the optical path (x/lambda). One more option is if the authors could include somehow the pointing vector analysis.

c) Line 82: “a ratio of radius to lattice constant (r/a) of 0.35”. The authors should define better the definition of the radius r, which is introduced at this point for the first time. The radius r could be for example shown somehow in the geometrical sketch in Fig.2

d) Line 119. “rt” is not defined.

e) The tolerance analysis is also missing (for example what happens if the material refractive index change with temperature). A brief analysis should be done or at least some considerations on this problem because the performance of PC etc..in terms of central wavelength, bandwidth etc.. definitively depend also on temperature by means of the optothermal coefficient dn/dT. A paper describing these effects for Photonic Band Gap to be included in the reference is for example:

[1] Larciprete M.C., et al,  J. Appl. Phys.  92, 2251 (2002).  DOI 10.1063/1.1499981

f) We guess that the authors should enrich the bibliography by citing some recent works on perfect absorbers and hot to optimize the design of PBG [2,3] should be cited in the reference list:

[2] R. Li Voti, ROMANIAN REPORTS IN PHYSICS (ISSN:1221-1451), 64, 446 (2012).

[3] R. Li Voti, Optimization of a perfect absorber multilayer structure by genetic algorithms, Journal of the European Optical Society-Rapid Publications 14, 11 (2018).

Author Response

We thank you for the careful and detailed review. We read the review in detail and revised our journal carefully.

a) Abstract.
 “With appropriate loss, we obtained zero reflection (CPL) in a single port by using a PMC-like boundary”. In the abstract the authors anticipate the main results of the numerical simulations, which are correctly reported in the conclusions with more details. The risk is that the abstract is not clear. We are wondering if the authors may re-write the abstracts just mentioning that in this paper they have studied, analyzed and optimize the structure in order to achieve a nearly zero reflection condition.
-> We modified the abstract
‘We numerically demonstrated single-port coherent perfect loss (CPL) with a Fabry-Perot resonator in a photonic crystal (PC) nanobeam by using a perfect magnetic conductor (PMC)-like boundary. The CPL mode with even symmetry can be reduced to a single-port CPL when a PMC boundary is applied. The boundary which acts like PMC boundary, PMC-like boundary, can be realized by adjusting the phase shift of the reflection from the PC when the wavelength of the light is within the photonic bandgap wavelength range. We designed and optimized simple Fabry-Perot resonator and coupler in nanobeam to get PMC-like boundary. To satisfy the loss condition in CPL, we controlled the coupling loss in the resonator by modifying the lattice constant of the PC used for coupling. By optimizing the coupling loss, we achieved zero reflection (CPL) in a single port with a PMC-like boundary.'

b) Figure 1 gives only a general idea of the process. Probably the reference to the real case with real materials etc.. could help. Moreover the “x arb” axis should be better replaced by the optical path (x/lambda). One more option is if the authors could include somehow the pointing vector analysis.
-> We already included in the Reference (Ref. [19]).  For the label in the x-axis in figure 1, we did not change it because one can get the wavelength information from the figure.

c) Line 82: “a ratio of radius to lattice constant (r/a) of 0.35”. The authors should define better the definition of the radius r, which is introduced at this point for the first time. The radius r could be for example shown somehow in the geometrical sketch in Fig.2
-> We added lattice constant ‘a_L','a_R’, hole radius ‘r_L’,'r_R], 'r_t' in figure 2 (a).
   Also, we changed the caption in figure 2. 
   'a_L' and 'a_R' are lattice constants of PC at left side and right side of resonator and 'r_L' and 'r_R' are radii of holes at left side and right side of resonator,  and 'r_t' is the radius of tapered hole.

d) Line 119. “rt” is not defined.
-> We added the radius of the tapered hole ‘r_t’ at figure 2 (a) and defined in caption of figure 2.

e) The tolerance analysis is also missing (for example what happens if the material refractive index change with temperature). A brief analysis should be done or at least some considerations on this problem because the performance of PC etc..in terms of central wavelength, bandwidth etc.. definitively depend also on temperature by means of the optothermal coefficient dn/dT. A paper describing these effects for Photonic Band Gap to be included in the reference is for example:
[1] Larciprete M.C., et al,  J. Appl. Phys.  92, 2251 (2002).  DOI 10.1063/1.1499981
-> According to ref. [35], the refractive index of silicon has variation about 0.01 at ±50 K difference from room temperature at wavelength of 1550 nm. Deviation of the refractive index of ± 0.01 is not a significant effect on the wavelength of the nanobeam resonator. Therefore, tolerance of temperature is not significant. We added sentence in paragraph 2 page 7 (Line 212). 
‘The photonic band gap is affected by temperature because the refractive index of material is varied with temperature [39]. However, the refractive index of silicon used in our simulation is robust from temperature. Refractive index of silicon is varied about ±0.01 at ±50 K difference from room temperature at the wavelength of 1550 nm [40], which makes deviation of resonance wavelength smaller than 5 nm.'

[39] Larciprete M.C., et al,  J. Appl. Phys.  92, 2251 (2002).  DOI 10.1063/1.1499981
[40] H. H. Li. Refractive index of silicon and germanium and its wavelength and temperature derivatives, J. Phys. Chem. Ref. Data 9, 561-658 (1993)

f) We guess that the authors should enrich the bibliography by citing some recent works on perfect absorbers and hot to optimize the design of PBG [2,3] should be cited in the reference list:
[2] R. Li Voti, ROMANIAN REPORTS IN PHYSICS (ISSN:1221-1451), 64, 446 (2012).
[3] R. Li Voti, Optimization of a perfect absorber multilayer structure by genetic algorithms, Journal of the European Optical Society-Rapid Publications 14, 11 (2018).
-> We added sentence and reference as [4] in paragraph 1 page 1 (Line 14).
‘In recent years, many studies have been implemented to increase the light absorption efficiency based on metal nanostructure, graphene and multilayer structure.’
And added reference as [22] in paragraph 4 page 2 (Line 61). 
‘A simple example of a PC is the Bragg reflector, which is a one-dimensional PC that can perfectly reflect incident light.'

Reviewer 3 Report

Results in this paper are very interesting. It can be published after addressing my comments:

The thickness of silicon is given for silicon-on-silica in the paper. 3D simulation is needed. Authors have reported the photonic bandgap. Is the result based on 2D or 3D simulation?

E-field intensity in xy-plane is reported. Authors should include the E-field intensity in xz-plane.

Author Response

We thank you for the careful and detailed review. We read the review in detail and revised our journal carefully.

The thickness of silicon is given for silicon-on-silica in the paper. 3D simulation is needed. Authors have reported the photonic bandgap. Is the result based on 2D or 3D simulation?
-> Our simulation is based on three-dimensional simulation. We modify sentence in paragraph 3 page 3 (Line 92).
'We calculated the photonic bandgap using the finite element method (COMSOL Multiphysics) in three-dimension' 
And also modify sentence in paragraph 1 page 4 (Line 98). 
'All simulations are performed in three-dimension.'

E-field intensity in xy-plane is reported. Authors should include the E-field intensity in xz-plane.
-> We added a figure of E_norm distribution in the xz-plane at Figure 2 (d).

Reviewer 4 Report

In this paper, author insist that the perfect magnetic conductor (PMC)-like boundary is realized by using one-dimensional photonic crystal and coherent perfect loss (CPL) is also achieved by using PMC-like boundary. However PMC-like boundary mentioned in this paper is just a perfect reflector with adjusted phase shift. So, this boundary works only at designed wavelength. I doubt this boundary is called PMC-like boundary. In addition, the origin of loss is not well described. At least propagation field should be shown. In this paper, material absorption loss is neglected. Therefore, the loss should be radiation loss toward lateral direction or out-of-plane direction. The relation between PMC-like boundary and CPL is also unclear. The theory related PMC-like boundary and CPL should be clearly described.

Author Response

We thank you for the careful and detailed review. We read the review in detail and revised our journal carefully.

In this paper, author insist that the perfect magnetic conductor (PMC)-like boundary is realized by using one-dimensional photonic crystal and coherent perfect loss (CPL) is also achieved by using PMC-like boundary. However PMC-like boundary mentioned in this paper is just a perfect reflector with adjusted phase shift. So, this boundary works only at designed wavelength. I doubt this boundary is called PMC-like boundary. In addition, the origin of loss is not well described. At least propagation field should be shown. In this paper, material absorption loss is neglected. Therefore, the loss should be radiation loss toward lateral direction or out-of-plane direction. The relation between PMC-like boundary and CPL is also unclear. The theory related PMC-like boundary and CPL should be clearly described.
-> Our goal of simulation is reducing the number of ports of CPA from 2 to 1. In the Fabry-Perot resonator, the resonance with a single port can be achieved using the symmetry. For the CPL with a symmetric mode, we need 2nπ (even symmetry) of phase difference and the same intensity between incident and reflection light. PMC boundary plays the same role. We designed and optimized this boundary using photonic crystal. Therefore, we named this boundary ‘PMC-like boundary’. We add sentence in paragraph 3 page 2 (Line 57).
‘The boundary which behaves as PMC or PEC can be realized with photonic crystals (PCs) on a chip. We named this boundary the PMC(PEC)-like boundary.'
In our simulation, we are using loss for CPL as scattering loss by scattering toward free space because materials used in our simulation do not have absorption. We mentioned it in paragraph 2 page 5 (Line 152).
'At the wavelength of 1550 nm, SiO2 and Si do not have optical absorption. However, the loss in terms of the CPL includes not only absorption but also coupling loss in the resonator.'
And we modified sentence in paragraph 2 page 5 (Line 152).
‘In our case, the loss in the CPL condition originates from the coupling loss at the PC boundary which is the scattering loss to free space.'

Round 2

Reviewer 4 Report

I do not think the perfect magnetic conductor (PMC) is an appropriate expression. Because the reflection is caused just by photonic band gap effect and phase of reflected wave is adjusted by adjusting air hole size.

I think the out-of-plane radiation should be shown. That is, not only top view but also side view of propagating field should be shown. If no reflection is realized by radiation from the cavity,  I think it is not so novel phenomena. However, it is OK to be accepted.